# Development and validation of a CCD-laser aerosol detective system for measuring the ambient aerosol phase function

Yuxuan Bian[1,2], Chunsheng Zhao[2], Wanyun Xu[3], Gang Zhao[2], Jiangchuan Tao[2], Ye Kuang[2]

[1]State Key Laboratory of Severe Weather, Chinese Academy of Meteorological Sciences, Beijing, 100081, China

[2]Department of Atmospheric and Oceanic Sciences, School of Physics, Peking University, Beijing, 100871, China

[3]State Key Laboratory of Severe Weather & Key Laboratory of Atmospheric Chemistry of CMA, Chinese Academy of Meteorological Sciences, Beijing, 100081, China

*Correspondence to*: Chunsheng Zhao (zcs@pku.edu.cn)

**Abstract.** Aerosol phase function represents the angular scattering property of aerosols, which is crucial for understanding the

climate effects of aerosols that have been identified as one of the largest uncertainties in the evaluation of radiative forcing. So far, there is a lack of instruments to measure the aerosol phase function directly and accurately in laboratory studies and in-situ measurements. A portable instrument with high angular range and resolution has been developed for the measurement of the phase function of ambient aerosols in this study. The charge-coupled device-laser aerosol detective system (CCD-LADS), which measures the aerosol phase function both across a relatively wide angular range of 10°-170° and at a high resolution of

0.1°. The system includes a continuous laser, two charge-coupled device cameras and the corresponding fisheye lenses. The CCD-LADS was validated by both a laboratory study and a field measurement. The comparison between the aerosol phase function retrieved from CCD-LADS and Mie-scattering model shows good agreement. Compared with the TSI polar nephelometer, CCD-LADS has the advantages of wider detection range and better stability.

## 1 Introduction

The climate effect of aerosol optical properties is one of the greatest uncertainties in our understanding about the climate change (Pachauri et al., 2014). Instruments such as the integrating nephelometer were often used to measure the aerosol scattering coefficient in laboratory studies and field campaigns (Anderson et al., 1996; Heintzenberg and Charlson, 1996; Ma et al., 2011; Müller et al., 2011; Tao et al., 2014). However, besides the total scattering coefficient, the distribution of aerosol scattering at different directions also has significant impact on the direct climate effect of aerosols (Kuang et al., 2015; Kuang

et al., 2016b). The aerosol phase function ($p(\theta)$) is defined to describe the angular distribution of the aerosol scattering intensity (Hulst, 1957). $p(\theta)$ is one of the important properties determining the contribution of aerosols to the radiative balance of the atmosphere (Andrews et al., 2006). Some parameters such as the asymmetry parameter and the hemispheric backscatter fraction estimated from $p(\theta)$ are of great importance to the retrieval of remote sensing measurements and in the simulation of atmospheric radiative transfer models (Muñoz et al., 2002). If the particle is assumed to be spherical, there is a

comprehensive theory named the Mie scattering theory to describe the characteristics of aerosol scattering, when the particle

size is in the same scale with the wavelength of scattering light (Bohren and Huffman, 2008). p(θ) can also be calculated with the size and complex refractive index of particles by the Mie theory (Kim et al., 2010).

In past years, different research groups have developed several versions of polar nephelometers to measure how the scattering intensities of aerosol particles, cloud droplets and ice crystals changes with scattering angle. Muñoz et al. (2001, 2010, 2011) mounted a photomultiplier tube (PMT) on a mechanical arm which can rotate around a point on the laser light path in the same plane with the laser beam to change the scattering angle of the signal captured by the PMT. Castagner and Bigio (2006, 2007) focused the light scattered at a single spot with different scattering angles to another single spot by using two parabolic reflectors next to the light path. A plane mirror was placed at that point to reflect the scattering signals with different angles to a PMT by rotation. These two styles of instruments measured the angular distribution of scattering signals by using the rotational mechanism. This design will lead to an obvious uncertainty because the signals were not measured simultaneously. Barkey et al. (2002, 2007) made the sample flow perpendicular and intersect with the light path. Then many PMTs were mounted around the point of intersection in the same plane with the laser beam to capture the scattering signal from different scattering angles. The signals with different scattering angles were measured at the same time with this design, however, the angular resolution which is limited to larger than 8° per point is relatively low because the PMTs cannot be mounted too close to each other. Curtis et al. (2007, 2008) used an elliptical mirror to reflect the scattering light to a charge-coupled device (CCD) detector for the detection of aerosol phase function. By using CCD as detector, this method can offer a better angle and time resolution at a wider range of scattering angles than the other methods above. It just needs one detector and there is no need to move the detector during the measurement. However, the structure of this design is too complicated to be used in field measurements.

Recently, McCrowey et al. (2013) developed a miniaturized polar nephelometer, which can be used in the in-situ measurement based on the techniques of Curtis et al. (2007) and can then be calibrated in the laboratory using polystyrene latex (PSL) standard particles. A comparison between the results measured from this instrument and calculated from a Mie model showed a good agreement. The detection range of this instrument is from 20 to 155° scattering angle. Besides these studies, the Aurora 4000 polar nephelometer (Ecotech Pty Ltd., Australia) is currently the only commercial instrument that can measure the aerosol phase function. This product has a structure similar to that of the integrating nephelometer, while a backscatter shutter that is able to be positioned at any angle between 10-90° is mounted in the cavity to help the nephelometer measuring the light scattering from that angle, through to 170°. The Aurora 4000 can just measure the aerosol phase function in a scattering angle range of 0-90° for dry aerosols. These two instruments can be used to measure the scattering phase function of dry aerosols in in-situ measurements.

In this paper, a novel instrument named charge-coupled device-laser aerosol detective system (CCD-LADS) based on the CCD imaging principle and the optical structure of the fisheye lens is developed to measure the ambient aerosol phase function in

the field measurement at a wider range of detection angles and a higher accuracy. The validation in both laboratory and field measurement shows the ability of the CCD-LADS to measure the aerosol phase function.

## 2 Instrumentation and Methodology

### 2.1 Design of instrument

The CCD-LADS includes several main components: a high power continuous laser emitter, two CCD cameras, optical filters and fisheye lenses. The laser and CCD cameras are mounted on tripods and controlled by a laptop. Each component is portable and on a scale of a few cubic decimetres.

The emitting system of the CCD-LADS is mainly built with a solid continuous laser emitter. Nd: YAG is used as the solid laser material as the wavelength of the emitter is 532 nm. The transverse mode is near TEM00. The M2 factor is less than 2.0

while the divergence of beam is less than 2.0 mrad. The diameter at the aperture is 3.0 mm. The power of the laser is 1 W. To change the polarization state of the laser from linear to circular, a quarter-wave plate was mounted in front of the laser emitter. During the exposure time (few minutes) of the image, the circular-polarization light can be assumed as unpolarised.

The receiving system of CCD-LADS has three main parts, the CCD cameras, the optical filters and the fisheye lenses. The SBIG model STF-8300 CCD imaging camera, which has the KAF-8300 CCD sensor (ON Semiconductor, Phoenix, AZ, USA)

is used. The area array (17.96*13.52 mm) of pixels has 8.3 million (3326*2504) effective pixels, while each pixel is a square 5.4 μm on a side. The exposure time is from 0.1 s to 1 h. The A/D converter is 16 bit. Due to its outstanding performance, this product is often used in astronomical measurements and also measurements in the other research areas (Coenen et al., 2015). The quantum efficiency of the CCD is about 55% at 532 nm, while the linearity error is about 10%. This camera has an air-cooling unit to control the temperature of CCD.

The fisheye lens (Sigma Corp., Japan) has a 10 mm focus length and a F2.8 aperture. When this lens is used with a Nikon camera, the field of view can be 180°. Because of the size of the CCD arrays, when this lens is used with the STF-8300 camera, the field of view is about 120°. The equisolid projection, which means that the solid angle of the object is directly proportional to the area on the CCD arrays, is used by this lens (Miyamoto, 1964). The modulation transfer function of the lens shows that, according to the size of the CCD sensor, the difference of the sensitivities from the centre to the corner is less than 5%

(http://www.sigma-photo.co.jp/english/lens/wide/10_28/#/data).

To filter out the background noise from the sky radiation, an optical filter (Thorlabs, Newton, NJ, USA) is mounted between the CCD camera and the lens. The filter has a 532±2 nm wavelength, and a 10±2 nm full width at half maximum, while the minimum transmission at the peak is 70%.

Figure 1(a) is the sketch map of the geometric relationship of CCD-LADS. The laser is emitted horizontally, while a beam

trap is used to receive the laser beam on the other side. Besides the laser beam, two CCD cameras with fisheye lenses are

installed at the same altitude with the laser to capture the scattering signal from the laser beam, while the directions of the cameras are forward and backward, respectively. With the mounted lens, there is a one-to-one correspondence between the image of the laser beam captured by CCDs and the laser beam object according to the principle of image formation by lenses. When two CCD cameras are used in this system, the detective angle can be expanded to 10-170°. The angle resolution can reach 0.1° per pixel. The scattering signal from 0-10° and 170-180° cannot be detected, because the signal to noise ratio is significantly lower than the value needed to estimate the quantities effectively.

To decrease the total area of the instrument, the distance between the CCD cameras and the laser beam should be less than 1m. So the CCD-LADS system covers an area 12 m long and 1m wide. When the instrument is set up, the first step to do is to measure the relative position of the CCD cameras, the laser beam and the laser emitter. From the geometric relationship shown in Figure 1, we can know that the light scattered at different position on the laser beam will be collected by different pixels on the CCD, so that the scattering light at different angles can be retrieved from the image captured by CCD. Due to the open path structure of the CCD-LADS, the background noise is much higher in daytime than in night time. Currently, the CCD-LADS system can just estimate the nocturnal aerosol scattering phase function.

## 2.2 Methodology

### 2.2.1 Data acquisition and pre-processing

The data acquisition of CCD-LADS is to obtain the angle-resolved scattering signals from images captured by two independent CCD systems, and then merge the signals. Firstly, the CCD-LADS is set up as shown in Figure 1(a). The geometric relationships among the CCDs, laser emitter and light trap are measured by tape. Then the scattering angle of laser in the image should be calibrated. The direction of the CCD cameras are adjusted to make sure that the image of laser go through the centre of the pixel arrays of CCD. By using a beam block the backscattering light is blocked from going into the CCD and the pixel related to the 90° scattering angle can be referred from the calibrated image (Figure 1(d)). Because of the equisolid projection is used by the lens, the distance from a point on the image on the CCD to the centre of the pixels can be calculated as $R = 2f \times \sin(\theta/2)$, where $\theta$ is the angle in rad between a point in the real world and the optical axis, which goes from the center of the image through the center of the lens, f is the focal length of the lens (Miyamoto, 1964). So the scattering angle, which is related to the centre of the image can be calculated by substituting the distance from the pixel related to the 90° scattering angle to the centre of pixels in the calibrated image into the equation of the equisolid projection. In this way, each pixel on the image of laser will be associated with a scattering angle.

At the beginning of the measurement, the CCDs are cooled down to -15°C to minimize the noise from dark current. Then a test image with a 10 s exposure time is captured to fix the exposure time of measurement by evaluating the signal intensity of this image. Generally, the maximum of the signal intensity is tuned to about $2^{14}$ because the limitation is $2^{16}$. If the maximum increased to the limitation in an image, the exposure time will also be changed in the next image automatically. The exposure

time of these two CCDs that were always about 5-60 seconds in the past observations and should be in complete accordance for the comparison. After the test image, a dark frame image is captured for each CCD by using a shutter in front of the lens. The dark current noise from the process and transmission of the signal can be subtracted during the procedure of image configuration by subtracting the dark frame image from the regular image (Figure 2(a)). The images are captured automatically

after these steps.

After image captured, the scattering light of the laser beam are separated from the background noise in the image as the follow steps. Firstly, the central axis of the scattering signals of laser beam is fitted in the program (the red line shown in Figure 2(b)). Then the intensities of image on the perpendicular of this central axis (the blue line shown in Figure 2(b)) are fitted with a normal distribution,

$$f(x) = I_0 + I \times \frac{1}{\sqrt{2\pi}\sigma} exp\left(-\frac{(x-\mu)^2}{2\sigma^2}\right) \tag{1}$$

where $I_0$ is the intensity of the background noise, $I$ is the intensity of the scattering signal of the laser beam related to one scattering angle, $x$ represents the distance between the pixel on the perpendicular and on the central axis of the scattering signals, $\sigma$ and $\mu$ are the fitting parameters of the normal distribution. Combining with the calibrated one-to-one correspondence between the image of laser and the scattering angle, the angle-resolved scattering signals is obtained with the

above steps of data acquisition.

When the angle-resolved signals from two CCDs are obtained, the change of signals with angles can be merged by following the steps below. Firstly, the minimum angle $\theta_1$ and maximum angle $\theta_2$ of the overlap angular region of signals from two CCDs are set as the boundary angle of data merging (shadow zone in Figure 3). $\theta_1$ and $\theta_2$ are always around 50° and 80°, respectively. In this region, a transform coefficient with scattering angles $T(\theta)$ is calculated,

$$T(\theta) = \frac{I_1(\theta)}{I_2(\theta)} \tag{2}$$

$I_1(\theta)$ is the signal with the scattering angle $\theta$ captured by the first CCD while $I_2(\theta)$ is that of the second CCD. The lifted signal $I_2'(\theta)$ can be calculated by multiplying $I_2(\theta)$ with the average of $T(\theta)$ (Figure 3). For the region where $\theta < \theta_1$ or $\theta > \theta_2$, the signal $I_1(\theta)$ or $I_2'(\theta)$ is used as the merged scattering signal $I(\theta)$, respectively. For the overlap region, a linear weighting average is done between $I_1(\theta)$ and $I_2'(\theta)$,

$$I(\theta) = \begin{cases} I_1(\theta), & \theta < \theta_1 \\ \frac{\theta_2-\theta}{\theta_2-\theta_1} \times I_1(\theta) + \frac{\theta-\theta_1}{\theta_2-\theta_1} \times I_2'(\theta), & \theta_1 \leq \theta \leq \theta_2 \\ I_2'(\theta), & \theta > \theta_2 \end{cases} \tag{3}$$

Using the method above, the merged signals with scattering angles $I(\theta)$ can be estimated.

**2.2.2 The retrieval algorithm to determine aerosol phase function**

Figure 4 shows the flow chart of the retrieval algorithm to determine $p(\theta)$ from CCD-LADS measurements. According to the geometric structure of the CCD-LADS, the echo equation of CCD-LADS can be figured firstly,

$$I(\theta) = N_0 \tau_Z \tau_R \beta(\theta) \tag{4}$$

where $\beta(\theta)$ is the scattering function of atmospheric air molecules and aerosols, $\tau_Z$ and $\tau_R$ are the transmittances on the optical paths of laser emitting and scattering respectively, $N_0$ is the calibration factor that depends on the optical efficiency of the instrument. Depending on the area that the CCD-LADS covers, the longest distance between CCD cameras and the laser beam is less than 8m. In this range, an assumption that $\tau_Z = \tau_R = 1$ can be established with a threshold that the visibility should be larger than 1.5 km. The correlation between the visibility and extinction coefficient $k_{ex}$ can be expressed as $k_{ex} = 3/visibility(km)$ (Chen et al., 2012) which means that the assumption can be established if $k_{ex}$ is smaller than 2 km[-1]. In some extreme pollution processes with high concentrations of both aerosol and gaseous pollutants (Ma et al., 2011; Xu et al., 2011), the scattering and absorption of aerosols and gases ($NO_2$ (Dixon, 1940), $O_3$ (Burrows et al., 1999), etc.) may lead to extreme $k_{ex}$ values. If the $k_{ex}$ is more than 2 km[-1], the assumption cannot be applied while the transmittance can calculated with the measurement of visibility. With the assumption, equation (4) can be transformed to $I(\theta) = N_0 \beta(\theta)$.

Scattering phase function $p(\theta)$ is the normalized angular distribution of the scattering function,

$$p(\theta) = \frac{4\pi\beta(\theta)}{\int_0^{180}\beta(\theta)d\theta} = \frac{4\pi I(\theta)}{\int_0^{180}I(\theta)d\theta} \tag{5}$$

So the scattering phase function can be calculated directly from $I(\theta)$ measured by CCD-LADS. If the scattering function of aerosols $\beta_{aero}(\theta)$ is known, the $p(\theta)$ can be calculated. Therefore, a retrieval algorithm is built to separate the scattering signals with angles into the scattering of aerosols and air molecules (shown in the dashed box in Figure 4).

As the first step, the scattering coefficient of air molecules at near surface level $k_{sc-air}$ is calculated with the density of atmosphere by a Rayleigh scattering model,

$$k_{sc-air} = \frac{8\pi^3(m^2-1)^2}{3n_{air}\lambda^4} \tag{6}$$

where $n_{air}$ is the number density of air molecules, which depends on the surface pressure and temperature measured by the weather station, $m$ is the index of refraction of atmosphere, which depends on $n_{air}$ and the wavelength of the laser $\lambda$. The hemispheric backscattering coefficient of air molecules $k_{bsc-air}$ is a half of $k_{sc-air}$ (Bohren and Huffman, 2008).

To resolve the ratio between the air molecules and the total hemispheric scattering $R_{air} = \frac{k_{bsc-air}}{k_{bsc-air}+k_{bsc-aero}}$, the hemispheric backscattering coefficient of aerosols $k_{bsc-aero}$ are measured with an intergrating nephelometer here.

To solve the intensity of the total hemispheric backscattering scattering signals $I_{bsc}$, the angle-resolved scattering signals should be integrated from 90° to 180° scattering angle. Because of the detective angular range of CCD-LADS is 10° - 170°, the angular truncation correction is necessary to resolve the hemispheric scattering intensity. For the backward angular truncation, the scattering intensity in that range is assumed to be equal to the scattering intensity at the largest scattering angle that CCD-LADS can measured. After the correction above, the corrected intensity $I'(\theta)$ is used to obtain $I_{bsc}$,

$$I_{bsc} = \int_0^{2\pi}\int_{\pi/2}^{\pi} I'(\theta)\sin\theta\, d\theta\, d\varphi \tag{7}$$

Then the angle-resolved scattering signals of air molecules can be calculated with a molecular phase function (Bohren and Huffman, 2008),

$$I_{air}(\theta) = \frac{3(1+\cos^2\theta)}{4} \times \frac{I_{bsc} \times R_{air}}{2\pi} \tag{8}$$

where $I_{air}(\theta)$ is the calculated angle-resolved scattering signals of air molecules. According to equation (4), the aerosol phase function $p_{aero}(\theta)$ can be estimated as,

$$p_{aero}(\theta) = \frac{4\pi(I(\theta)-I_{air}(\theta))}{\int_0^{180}(I(\theta)-I_{air}(\theta))d\theta} \tag{9}$$

### 2.2.3 Error analysis

Two types of uncertainties determine the error of the retrieved aerosol phase function: the measurement errors caused by the processes of obtaining the angle-resolved signals, and an error introduced by the retrieval algorithm.

There are two sources of measurement errors in the data acquisition processes introduced in Section 2.2.1. Firstly, the measurement error of CCD used in the CCD-LADS is 10% according to the related manual. The relative difference between the fitted normal distribution introduced in equation (1) and the measured signal in the laboratory study is 8.8% ±1.5%, which can also certify the 10% measuring error on $I$ introduced by the manual of CCD. Secondly, the measurement of the geometric relationship will lead to at most 5% relative error on the scattering angle $\theta$ introduced by the resolution and accuracy of the used tools.

The relative errors on the merged angle-resolved signals $I(\theta)$ can be derived by applying a standard propagation of errors to equation (3) (Bevington and Robinson, 2003),

$$\left(\frac{\Delta I}{I}\right)^2 = F_{I_1}\left(\frac{\Delta I_1}{I_1}\right)^2 + F_{I_2'}\left(\frac{\Delta I_2'}{I_2'}\right)^2 + F_\theta\left(\frac{\Delta\theta}{\theta}\right)^2 \tag{10}$$

where $\sigma$ symbol means the standard deviation of variables, $\frac{\Delta x}{x}$ is equal to the relative error of $x$ and the propagation factor $F_x$ are defined as $F_x = \left(\frac{x}{I}\frac{\partial I}{\partial x}\right)^2$. By substituting the relative errors and the average signals into equation (10), the uncertainties on $I(\theta)$ are calculated as a distribution with angular shown in Figure 5. The values of uncertainties on $I(\theta)$ are between 10% and 19%, and varied with angles.

The uncertainties of the retrieval algorithm are introduced by the uncertainties of the input parameters. There are three groups of input parameters in the retrieval algorithm: merged angle-resolved signals, aerosol hemi-backscattering coefficient and temperature/pressure. The errors of the temperature and pressure are about 0.1 K and 0.1 hPa (Box and Steffen, 2001), respectively, which will lead to a 0.02% uncertainty on $k_{bsc-air}$. Combined the 10% uncertainties on the measured $k_{bsc-aero}$ (Heintzenberg et al., 2006), the uncertainty of $R_{air}$ can be calculated as 7% with the algorithm in Section 2.2.2. According to the algorithm shown in Figure 4, the uncertainty of the retrieved aerosol phase function are mainly dominated by the uncertainties of the merged signal shown in Figure 5, and also influenced by the uncertainty of $R_{air}$ in a way.

**3 Results**

**3.1 Laboratory Results**

To validate the ability of the CCD-LADS to measure the aerosol phase function, an indoor experiment was held in the laboratory in the Physics Building at Peking University during November 7-8th, 2015. The time resolution of CCD-LADS was set to 60 s during the experiment, while the angular detection ranged from 10 to 170°. The aerosol scattering coefficient, number size distribution, mass concentration of black carbon particles, ambient temperature and relative humidity were measured with an integrating nephelometer (Model 3563, TSI, Inc., Shoreview, MN, USA), a scanned mobility particle sizer (SMPS; Model 3936, TSI, Inc., Shoreview, MN, USA), an aerodynamic particle sizer (APS; Model 3321, TSI, Inc., Shoreview, MN, USA), a micro-aethalometer (Model AE51, Magee Scientific, Berkeley, CA, USA) and a dew-point chilled mirror sensor (Edgetech DewMaster), respectively.

Figure 6 shows the time series of several quantities during the laboratory experiment. The scattering/hemispheric backscattering coefficient of aerosols at 525 nm wavelength shown in Figure 6(b) and the mass concentration of black carbon particles shown in Figure 6(c) reveal the same pattern that first declines and climbs up afterwards. The same pattern can be discovered in the time series of particle number size distributions shown in Figure 6(d). The variation reflects the slow exchange between the air indoor and outdoor. The peak diameter of aerosol number size distribution was still around 100 nm, while it had a slight shift during the experiment. Therefore, the fine particles are dominant in the laboratory. The single scattering albedo (SSA) shown in Figure 6(c) was around 0.85 which means that the black carbon aerosol took up a relatively large proportion among the aerosol species, resulting in strong particle light absorption ability.

Combining the particle number size distributions measured with SMPS/APS and the mass concentration of black carbon aerosols measured with AE51 (Figure 6) into a modified Mie-scattering model, the aerosol optical properties including the aerosol phase function could be modelled (Ma et al., 2011). In this study (both laboratory and field study), the refractive index used for black carbon component is 1.95-0.79i (Seinfeld and Pandis, 2006), and for non-absorbing component is 1.53-10$^{-7}$i (Wex et al., 2002). The mass ratio between two different mixing states (external or core-shell, which are two different ways black carbon and non-absorbing aerosols are mixed) of black carbon aerosols is assumed to be 1:1 according to the result of Ma et al. (2012). Figure 7 shows the comparison among the aerosol phase functions retrieved with the CCD-LADS retrieval algorithm, modelled with the modified Mie model and offered by the aerosol classification from the Cloud-Aerosol Lidar and Infrared Pathfinder Satellite Observations (CALIPSO) aerosol products (Omar et al., 2009). The CALIPSO aerosol classifications are based on the cluster analysis of the Aerosol Robotic Network (AERONET) measurements to determine characteristic aerosol types (Omar et al., 2005). Here the red solid line shows the retrieved $p(\theta)$ from the CCD-LADS measurements with the retrieval algorithm introduced in Sect. 2.2.2, while the brown dashed line shows the retrieved $p(\theta)$ from CCD-LADS directly without considering the air molecules scattering influence. The blue dashed line shows the modelled

result, and the other dotted lines express the aerosol phase functions of different aerosol types from CALIPSO aerosol classification. The uncertainty of the retrieved $p(\theta)$ and simulated $p(\theta)$ with Mie model, which is about 30% (Ma et al., 2011), are shown as error bars. The result shows that the comparison between the modelled $p(\theta)$ and the $p(\theta)$ retrieved with the retrieval algorithm shows a better agreement than the comparison between the modelled $p(\theta)$ and the $p(\theta)$ retrieved

from the CCD-LADS measurements directly, especially for the backward scattering. The reason of this phenomenon is that the scattering coefficients of aerosols and air molecules are closer to each other for the backward scatter than for the forward scatter based on the background that the total scattering coefficient of aerosols is always much higher than that of air molecules. The comparison also shows that the retrieved $p(\theta)$ is closer to the aerosol phase function of the "biomass burning" aerosol among the six aerosol types classified from CALIPSO aerosol products. Compared with the other aerosol types, the "biomass

burning" aerosol represents a better absorption ability due to the larger percentage of black carbon aerosol and organic aerosol, and also a smaller effective diameter around 100 nm (Omar et al., 2005; Rissler et al., 2006; Zhu et al., 2017). The SSA and particle number size distribution of aerosols during the experiment shown in Figure 6 also have the similar characteristics with the "biomass burning" aerosol.

To further validate the quality of the retrieved result from the CCD-LADS measurement, a comparison was also carried out

among the $p(\theta)$ at 42° scattering angle resolved with different methods (Figure 8). The $p(\theta)$ at 42° scattering angle is relatively typical and comparable because 42° is the scattering angle used in the forward scattering visibility sensor (Kessner et al., 2013). The result of the comparison shows that the $p(\theta)$ from CCD-LADS measurement and Mie model have the same pattern and the average difference in the absolute values between these two $p(\theta)$ is less than 10%.

### 3.2 Field Measurements

During January 2016, a comprehensive field campaign focused on air pollution in winter was conducted at the roof of a school building in Yanqi campus of the University of Chinese Academy of sciences (UCAS) in Huairou district, Beijing (40°24′ N, 116°40′ E, 91 m a.s.l.). The observatory is 60 km away from the downtown of Beijing and is at the edge of the North China Plain (NCP), which makes it suitable for measuring the regional pollution properties of the NCP (Ma et al., 2016). During the campaign, all the instruments except for the CCD-LADS were housed in a laboratory with a steady room temperature as 20°C.

The aerosols were sampled from an inlet 5 m higher than the ground and then dried to a relative humidity less than 30% before flowing into the laboratory to measure the aerosol number size distribution, scattering coefficient, phase function and the mass concentration of black carbon aerosols at a dry condition. The CCD-LADS was mounted outside the laboratory at the same altitude to measure the scattering phase function of ambient aerosols. Depending on the limitation of the ambient condition, the angular detection range of the CCD-LADS was 30-160° in this campaign.

During the field measurement, the scattering phase function of dry aerosols could be resolved from two ways: Aurora 4000 polar nephelometer measurements and the modified Mie-scattering model with the related aerosol measurements. Under high

relative humidity condition, aerosol particles will absorb moisture in the atmosphere and exhibit hygroscopic growth significantly (Bian et al., 2014; Chen et al., 2014; Kuang et al., 2016a), and hence the scattering properties of ambient and dry aerosols are totally different. Therefore, the data collected at a relative humidity above 70% were eliminated from the comparison among the scattering phase functions of dry and ambient aerosols obtained by different methods. Figure 9 shows the result of the comparison mentioned above. The results from three methods are consistent with one another in the overlap of the detectable scattering angular range. Compared with the other results, the retrieval of CCD-LADS measurement enhances the backward scattering fraction of aerosol. This might be caused by the angular range (30-160°), which did not reach 10-170° and therefore might have increased errors in retrieving the angular distribution of aerosol scattering. The $p(\theta)$ from Aurora 4000 measurements have the similar average pattern with the results from other methods, but the deviation of its pattern is obvious. Compared to the Aurora 4000 results, there are two significant advantages of CCD-LADS: wider detection range and better stability.

## 4 Discussions and Conclusions

A novel instrument named charge-coupled device-laser aerosol detective system (CCD-LADS) was developed to measure the nocturnal ambient aerosol phase function in the ambient atmosphere at a wider range of detection angles and a higher accuracy. The validation in both laboratory and field measurement shows the ability of CCD-LADS to measure the aerosol phase function. A laser is emitted horizontally, while two CCD cameras with fisheye lenses are installed besides the laser beam at the same altitude to capture the scattering signal from the laser beam with the cameras facing forward and backward, respectively. Then the signal captured by the two cameras are merged into one signal curve. The detectable angular range is from 10-170°, while the angle resolution reach 0.1° per pixel. A retrieval algorithm is developed to subtract the influence of air molecules scattering with the integrating nephelometer and weather station measurements. The uncertainties of CCD-LADS were discussed.

To validate the ability of CCD-LADS to measure the aerosol phase function, an indoor experiment was held in the laboratory of the Physics Building at Peking University during November 7-8th, 2015. During the experiment, the angular detection range was from 10-170°. The comparison between the modelled $p(\theta)$ and the retrieved $p(\theta)$ shows an excellent agreement. Both of them are close to the aerosol phase function of the "biomass burning" aerosol from CALIPSO aerosol products. The comparison result is reasonable, because the SSA and particle number size distribution of aerosols during the experiment also had similar characteristics with the "biomass burning" aerosol. The comparison of the $p(\theta)$ at 42° scattering angle acquired by different methods also shows good agreements on both patterns and absolute values.

During January 2016, a comprehensive field campaign focused on air pollution in winter was organized at the roof of a school building in Yanqi campus of the UCAS. Depending on the limitation of ambient condition, the angular detection range of the CCD-LADS was 30-160° in this campaign. The retrieved aerosol phase function with CCD-LADS measurements is consistent

with both the Aurora 4000 measurement and the modified Mie model results in the overlap region of the detectable scattering angular range. Compared with the Aurora 4000 measurements during this campaign, the CCD-LADS measurements are steadier.

Both the laboratory experiment and the field measurement have demonstrated that the CCD-LADS is a robust instrument, fully capable of measuring the ambient aerosol phase function under different conditions. Overall, compared with the laboratory-scale instruments, the CCD-LADS measured aerosol phase functions in a wider angular range and a higher angular resolution.

## 5 Data availability

The averaged retrieved aerosol phase function used to create Figure 7 is attached in Supplement. The CALIPSO aerosol classification data is listed in the reference. The whole data set can be accessed by request to the corresponding author at zcs@pku.edu.cn.

## Acknowledgements

This work is supported by the National Natural Science Foundation of China (41590872, 41375134).

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

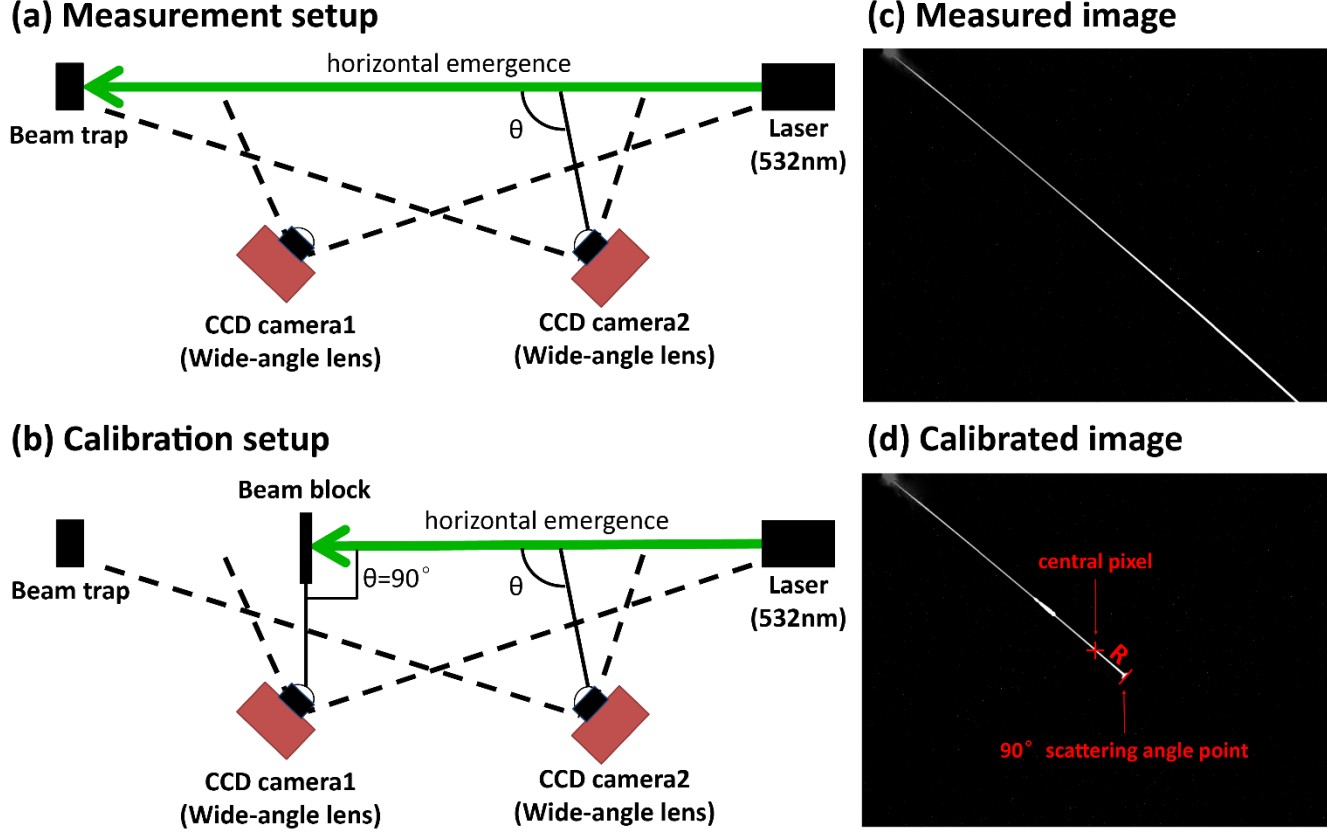

Figure 1: Sketch map of the geometric relationship and the sampling image of CCD-LADS

## (a) Dark current noise subtraction

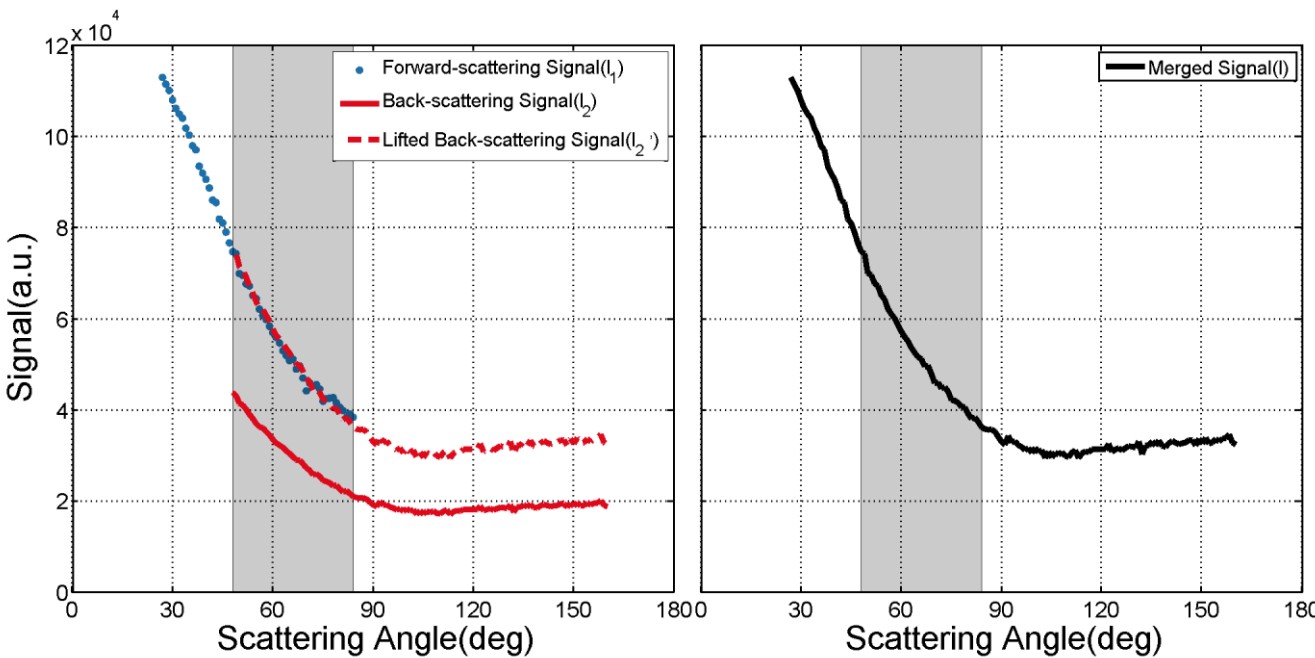

## (b) Background noise subtraction

**Figure 2: Noise subtraction of CCD-LADS: (a) Dark current noise subtraction; (b) Background noise subtraction.**

**Figure 3: Signal merging of two CCD cameras. Besides the signals captured by the first CCD (blue dotted line) and the second CCD (red solid line), the lifted signal from the second CCD (red dashed line) is also shown in the left drawing. The merged signal is shown in the right drawing.**

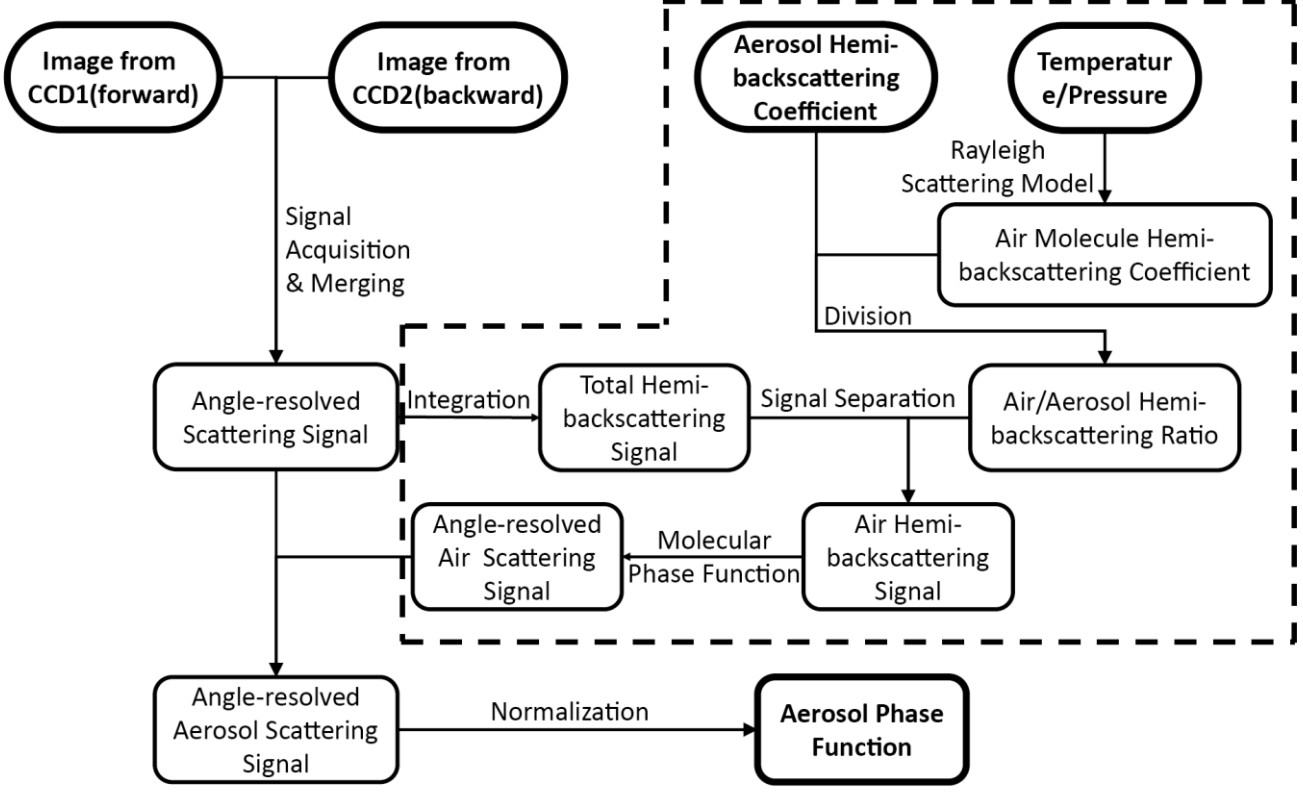

**Figure 4: Flow chart of the retrieval algorithm to determine aerosol phase function from CCD-LADS measurements (the processes in the dashed box is used to subtract the scattering signal of air molecules from the total scattering signal)**

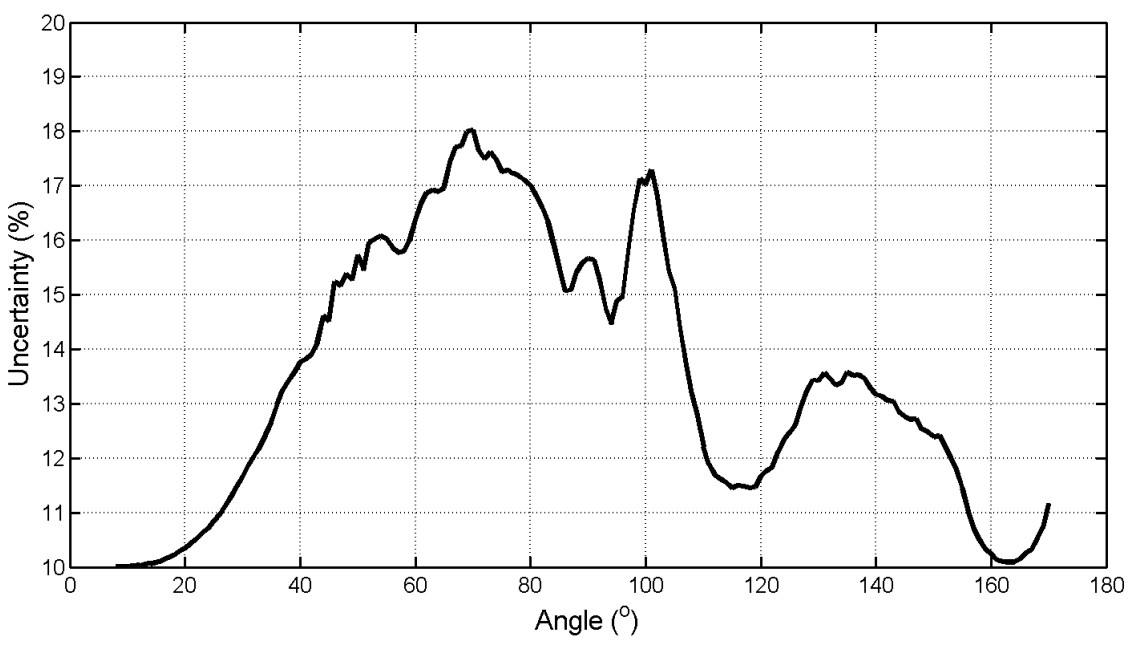

5    **Figure 5: Uncertainties of the merged angle-resolved signal from CCD-LADS measurement**

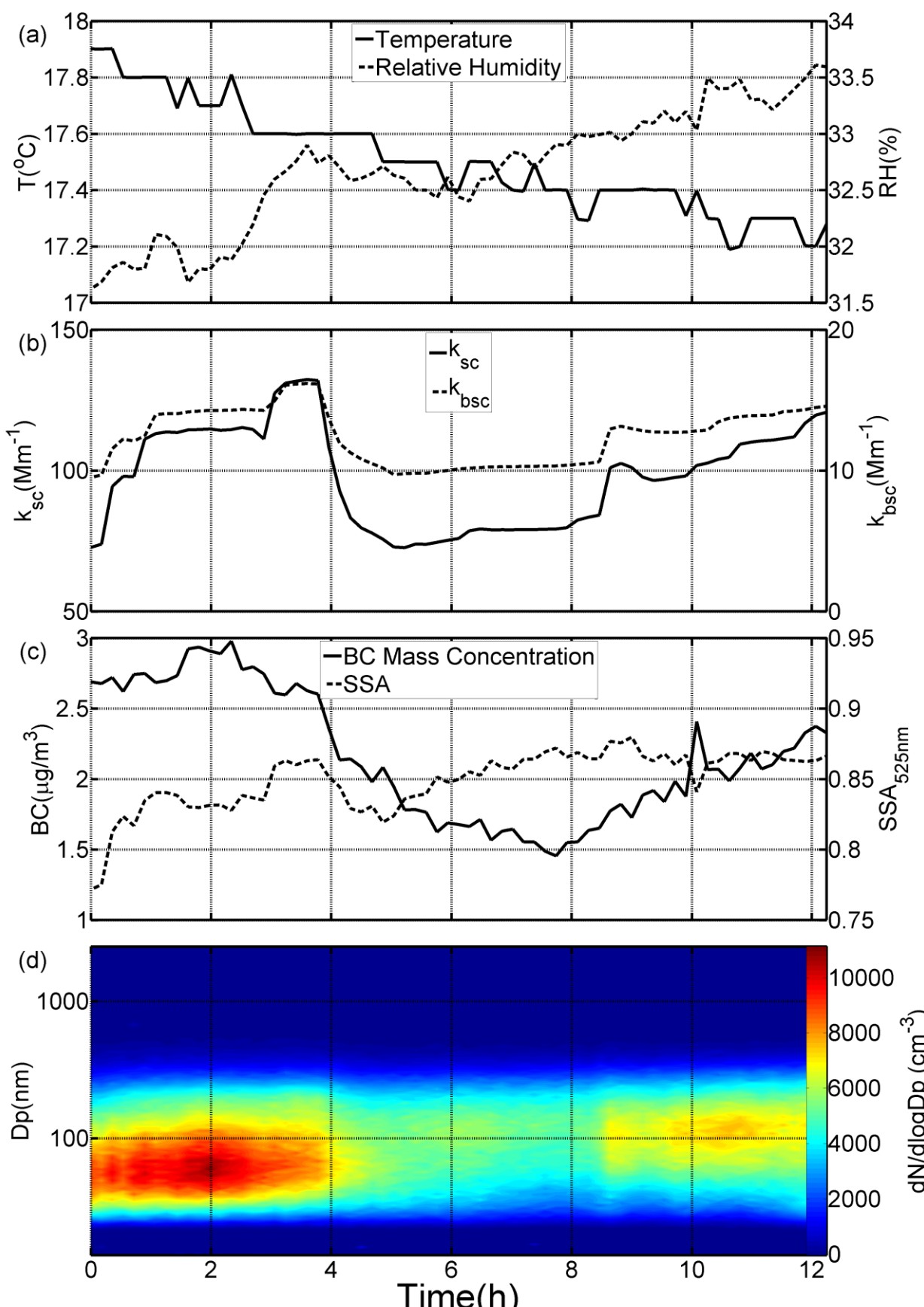

**Figure 6: Time series (a) temperature (solid line) and relative humidity (dashed line) in the laboratory, (b) scattering coefficient (solid line) and hemispheric backscattering coefficient (dashed line) of aerosols at 525nm wavelength, (c) mass concentration of black carbon particles (solid line) and single scattering albedo of aerosols at 525nm wavelength (dashed line), (d) PNSD of aerosols during the laboratory study at Peking University in 2015.**

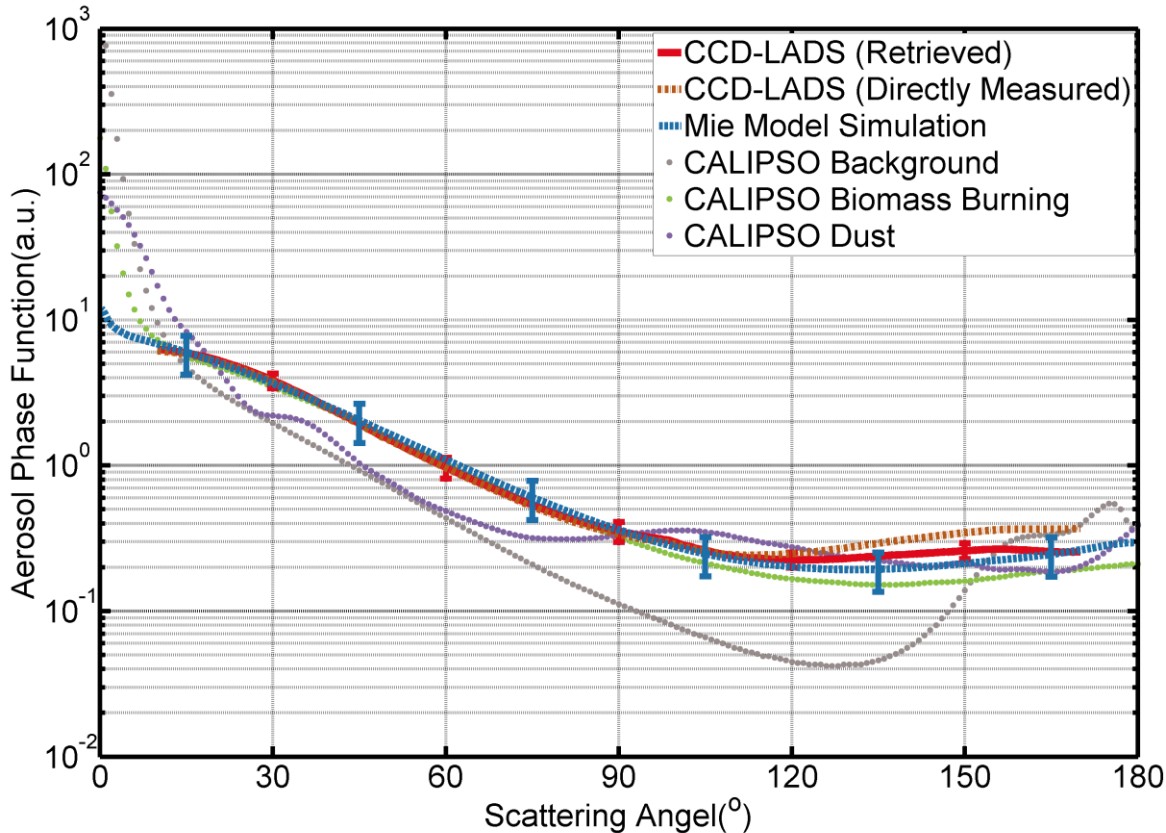

**Figure 7: Comparison between aerosol phase function obtained from CCD-LADS measurements (red solid line shows the result estimated with the retrieval algorithm, brown dashed line shows that estimated directly with the measurements), modelled with modified Mie model (blue dashed line) and offered by previous studies with CALIPSO (different colors of dotted lines represent different aerosol types).**

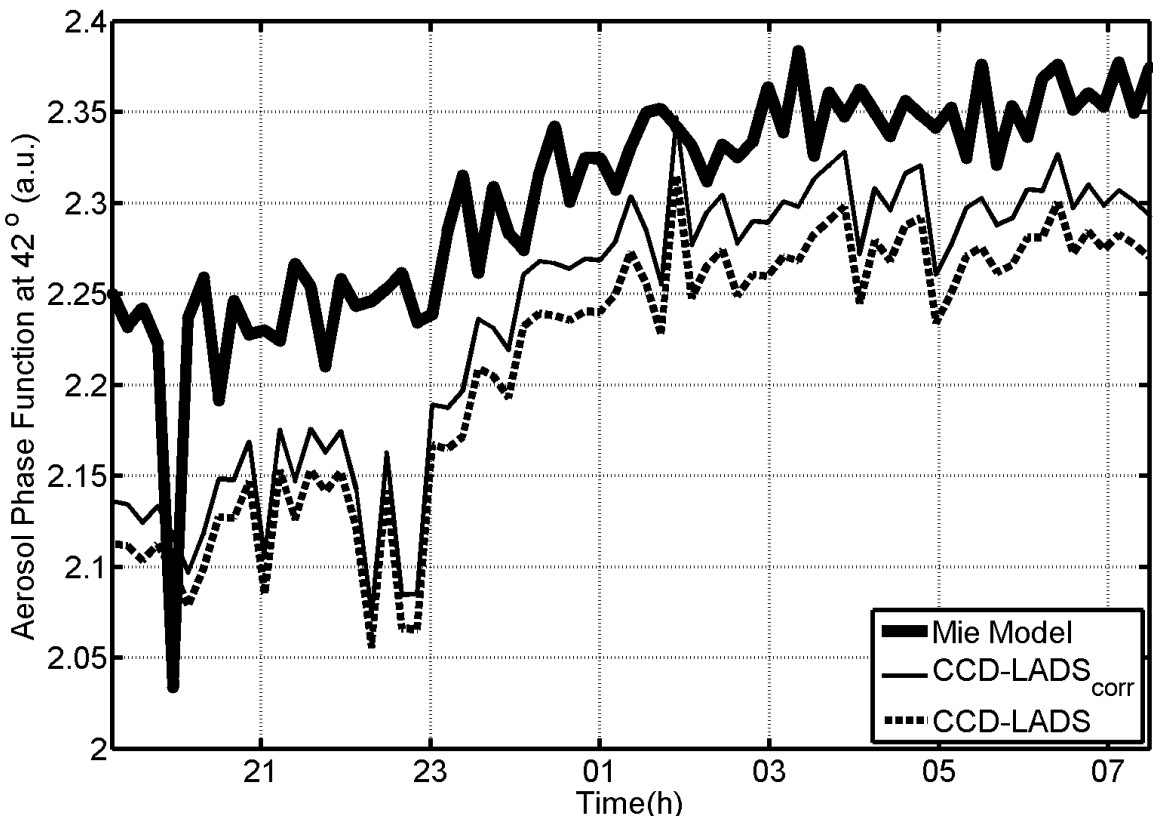

**Figure 8: Comparison between aerosol phase function at 42° scattering angle obtained from CCD-LADS measurements (the results estimated with the retrieval algorithm are shown with fine solid line, while the values estimated directly with the measurements are shown with dashed line) and modelled with modified Mie model (shown with bold solid line).**

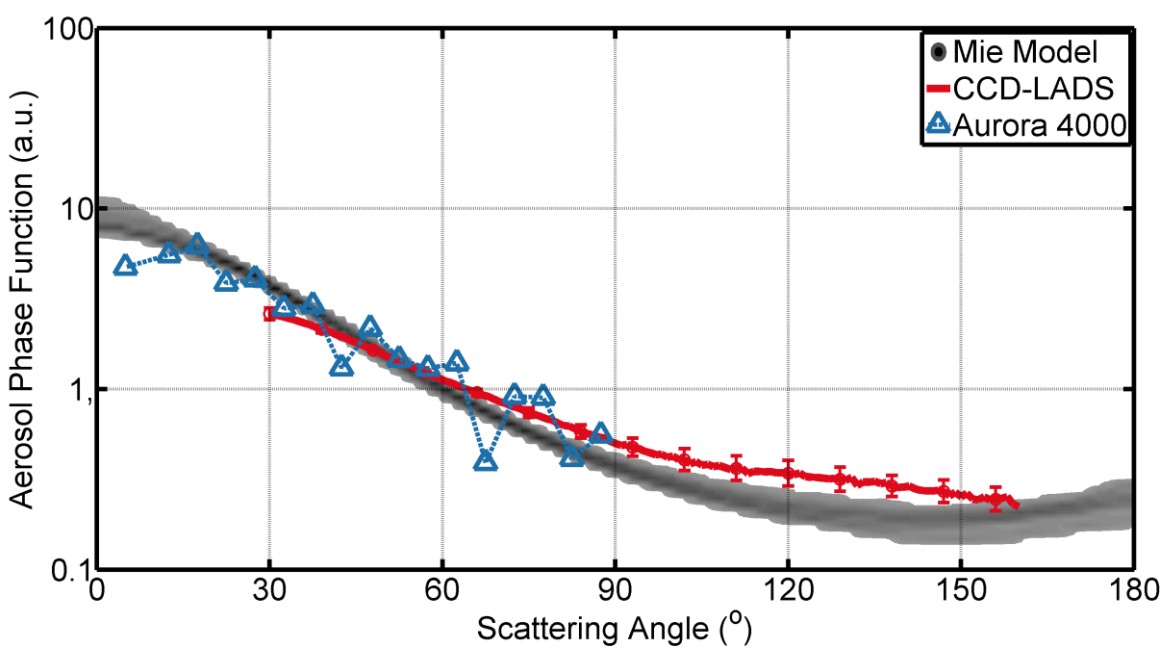

**Figure 9: Comparison between aerosol phase function retrieved from CCD-LADS measurements (red line shows the average value, the error bar shows the standard deviation), measured from Aurora 4000 polar nephelometer (blue triangle) and modelled with modified Mie model (grayscale map).**