# Peer review of "Development and validation of a CCD-laser aerosol detective system for measuring the ambient aerosol phase function"

_Atmospheric Measurement Techniques, 2016_

## Referee Comment (RC1) · Anonymous Referee #1 · 7 Mar 2017

General Comments:

This paper describes a newly developed CCD and laser-based detection system for measuring aerosol phase function.

Overall, this instrument has some advantages, such as a very simple setup and design. I believe that, if the efficacy can be shown, the idea for this instrument is a significant advance and falls under the scope of AMT.

However, I feel that there are some outstanding questions that need to be addressed before the paper can be published. These are listed below in the specific comments.

I believe that if these questions are addressed, then the paper may be publishable.

[Figure]

However, the following questions are crucial to the measurement and the authors may need to perform more measurements to ensure the efficacy of the instrument.

In general, the English in the paper is relatively unclear and needs to be edited.

Specific Comments:

1) What is the polarization state of the laser relative to the plane of scattering?

Although the authors do not specifically address this, I suspect that they have taken this detail into consideration. The point of my comment is that the phase function of the aerosol depends on the incoming polarization of the laser relative to the plane of scattering. The laser polarization can be oriented perpendicular to, parallel to, or somewhere in between these two positions. In order to compare the measurements to Mie theory, the polarization state must be included in the Mie calculation. This is not mentioned in the paper.

The scattered light will also be polarized and this needs to be addressed as well. For example, the detectors may detect different polarizations of light slightly differently. A flat-field calibration across the CCDs is also needed to verify that different angles of incident light to the detector are measured with equal sensitivity.

2) How can the authors be sure that they are not detecting scattered light from multiple points along the laser into their CCD detectors? Although this is mentioned in the text, it is not clear that this is true.

In Figure 1, the authors show the setup, including scattering angle into the detector. But the aerosol is not constrained in any way along the path of the laser. Therefore, light scattering from a particle near the laser and from a particle at a different point along the laser beam could strike the detector at the same location.

It would be highly useful to show some raw data from the two CCD detectors for one measurement to show the relationship between the two and how the data is spliced together. At what scattering angles do the two CCD measurements overlap?

How do the authors determine the scattering angle that corresponds to each pixel position on the CCD? Is there some sort of measurement of a known scatterer that could be used to calibrate the angle, such as PSL or ammonium sulfate?

In general, it would be a good idea to perform the measurement with a known compound to better determine the efficacy of the instrument.

3) More detail is needed regarding many aspects of the study. There needs to be more information regarding how the Mie calculations were performed. What was the refractive index used? Were the size distributions from the SMPS? 4) The instrument will measure scattered sunlight when used during daylight hours. Using a 532 nm filter in front of the cameras will block much of the scattered sunlight, but the sunlight near 532 nm will be scattered by the particles and will reach the detector. Was there any attempt to block the sun from reaching the particles? For example, could a dark curtain or sampling box be used? 5) Fluctuating temperature in the ambient measurement will likely cause fluctuations in the laser alignment during the measurements. Also, temperature fluctuations could affect the CCD efficiencies. Were the CCD cameras electronically cooled or was the temperature stabilized in any way? 6) The authors claim that a significant portion of the particles measured were biomass burning particles. These particles could absorb a significant fraction of light. Was the absorption considered in the Mie calculations or was only scattering considered? 7) Ambient gases, such as nitrogen dioxide and ozone could absorb a significant fraction of light. It would be useful on Page 4, Line 20, for example, to discuss more detail in the assumption that the transmittance of the laser is 1. 8) The entire discussion of the retrieval algorithm is not clear (Section 2.2.2). More detail is needed. Please describe exactly how the retrieval algorithm works. Why is the retrieval needed? You are measuring the phase function, correct? Why would you need to retrieve it? 9) Error bars (uncertainties) of the measured phase functions are needed to better understand the overlap between the models and measurements (in Figure 4, for example).

Technical Corrections:

In general, there are not enough citations to the literature in the introduction. For example, Bohren and Huffman should be cited for Mie theory. The IPCC report should be cited for the first sentence (Page 1, Line 20). There are more field measurements that used an integrating nephelometer. Overall, more citations are needed.

Throughout the paper, "figure" should be capitalized as "Figure 1".

N0 and T are both used as symbols for transform coefficient (Page 4, Lines 6 and 19). How are these different?

Page 1, Line 24 should read "The aerosol phase function. . .".

Page 2, Line 5 should read "A plane mirror was placed at that point and reflected the scattering signals. . ."

Page 2, Line 7: Is "linearity" correct? If so, please explain.

Page 2, Line 28. "mainly" is not needed. More information is needed about the laser. The make, model, etc. is needed.

Page 3, Line 20 should read "The CCD-LADS system covers an area 12 m long and 1 m wide."

Page 3, Line 21. How is the background noise measured? It would be useful to show a plot of the signals measured during the daytime and nighttime. Also, why is the noise so much higher in the daytime? See the comment above regarding the filters above. I suspect that you have a large portion of ambient light being scattered by the particles. Depending on the angle of the sun, this could affect the ambient measurements.

Page 4, Lines 8-13. More detail is needed about the linear weighting and the data processing in general.

Page. 4, Line 30. The authors state that the hemispheric backscattering is half of the full scattering. This is true for Rayleigh scattering, but it depends on the polarization of the light. Bohren and Huffman, or a Rayleigh scattering calculation needs to be cited

here.

Page 5, Lines 1-9. This section is very confusing. Was a separate measurement of the air molecule scattering measured by an integrating nephelometer during this study? More detail is needed. What is meant by "the percentage of air scattering", for example? See the comment above.

Page 6, Line 13. What is the external/core-shell ratio of 1:1? What does this mean? 1:1 core and shell ratio by volume?

Page 6, Line 23. The explanation that the scattering abilities of aerosol particles and gas molecules match better in the back scatter than the forward scatter is not clear. In the phase function, the Rayleigh and Mie scattering generally match in the forward and reverse directions (0 and 180 degrees).

Page 6, Line 25 and Figure 4. Biomass burning aerosol is not shown on the graph. Also, this needs a reference for the discussion about biomass burning size, absorption, etc.

Page 7, Line 2. The authors state that the difference "Is not obvious", but there is clearly a difference between the Mie model and the measurement in Figure 5. This is not clear.

Page 7, Line 13. Why was the instrument limited to 30-160 degrees because of the ambient conditions?

Page 7, Line 16. It is more correct to say that the particles will exhibit hygroscopic growth rather than "grow up". Also in Line 18, why was the data collected at RH > 70% eliminated? How was 70% RH chosen? Many particles exhibit water uptake at lower relative humidities. Biomass burning aerosol is highly hygroscopic and shows continuous growth. 70% seems to be arbitrarily chosen.

Page 7, Line 18. It is better to say "eliminated" than "kicked off".

Page 7, Line 26. What is meant by better stability? Was the data less noisy? This is not clear.

Page 8, Line 18. What is meant by "steadier"?

Figure 3: The data shown here was not collected by the CCD-LADS instrument that is the focus of this paper. Why is this data shown? Was it used in the Mie calculation to compare to the CCD-LADS measurements?

---

## Referee Comment (RC2) · Anonymous Referee #2 · 12 Mar 2017

The manuscript presents a system for measurement of aerosol scattering phase function at ambient conditions. The principle and design of the system is briefly described. Results of lab and field test are also shown and compared with simulation and other measurements. The performance of the system seems to be very good.

I think it is a good idea, since the design is simple, easy to be applied, and would not cost much. Therefore I would recommend the publication of this manuscript in AMT if the questions below are well addressed.

Major comments:

1. The description of the data processing algorithm (section 2.2) is unclear and hard to

follow. Especially for section 2.2.2, I did not really get how it works. Also, some information is not mentioned. For example, what is the time resolution of the measurement? Does it depend on the aerosol concentration? I suggest the author to re-organize section 2.2, providing more details, describing the data processing algorithm step by step, in a more logical way.

2. The evaluation of the measurement uncertainty of the presented system is missing in current manuscript. There are many possible sources which may add uncertainties on the phase function provided by the new system. For example, the relative angle between the laser beam and the optic axis of the camera might be not exactly as what you expected. Also, there is always ambient light influencing the signal. I suggest the author to add a new section discussing about all those possible uncertainty sources, and give an overall estimate of the measurement uncertainty.

3. the English language needs to be improved.

Minor comments:

Section 1: The background at the beginning of section 1 is very weak. I suggest the author to write a bit more about why the phase function is important.

In section 1 you listed many methods which can provide aerosol phase function. Are they widely applied? If not, why? What are the advantage and disadvantage of those methods? One can give very short comments on each method.

Last para in section 1: it is better to start with "in this paper, we propose...". Otherwise it seems you are still talking about previous studies.

P3L17: the detective angle range can be expanded to 10-170.

P3L20: how do you measure the direction of the two CCD cameras? How do you ensure that they are pointing to the right direction?

P4L1: what is the typical exposure time?

P4L2: I do not understand "the central axis of the signals from the scattering light is fitted in the program". Explain it in detail or just remove it if it is not important for audience to understand the system.

Fig2: signal merging is missing in this flowchart.

P4L21: in extreme cases, e.g. heavy haze or fog, what is the uncertainty of assuming a tau of 1? Maybe one can give a threshold of visibility above which the uncertainty is negligible.

Section 3: Mie calculation was used to simulate the phase function at dry condition. Can you give an estimate of the uncertainty of the simulation? And I suggest the author to mark the uncertainties of measurement and simulation as error bars.

P6L25: I did not see "biomass burning" in figure 4.

---

## Author Comment (AC1) · 2 May 2017

The responses and the revised manuscript are included in the supplement zip file.

Please also note the supplement to this comment:
http://www.atmos-meas-tech-discuss.net/amt-2016-390/amt-2016-390-AC1-supplement.zip

---

## Author Response (AR1)

Dear Dr. Wagner,

We greatly appreciate the referees for their valuable comments and suggestions. The manuscript has been revised accordingly. Point-by-point responses to all the comments along with the revised manuscript and figures have been uploaded.

Best Regards
Yuxuan Bian and co-authors

**Response to Anonymous Referee #1:**

**General comments:**

*This paper describes a newly developed CCD and laser-based detection system for measuring aerosol phase function.*

*Overall, this instrument has some advantages, such as a very simple setup and design. I believe that, if the efficacy can be shown, the idea for this instrument is a significant advance and falls under the scope of AMT.*

*However, I feel that there are some outstanding questions that need to be addressed before the paper can be published. These are listed below in the specific comments.*

*I believe that if these questions are addressed, then the paper may be publishable.*

*However, the following questions are crucial to the measurement and the authors may need to perform more measurements to ensure the efficacy of the instrument.*

*In general, the English in the paper is relatively unclear and needs to be edited.*

**Response:** Thanks for the comments. The English has been improved in the revised manuscript.

**Specific comments:**

*1. What is the polarization state of the laser relative to the plane of scattering?*

*Although the authors do not specifically address this, I suspect that they have taken this detail into consideration. The point of my comment is that the phase function of the aerosol depends on the incoming polarization of the*

*laser relative to the plane of scattering. The laser polarization can be oriented perpendicular to, parallel to, or somewhere in between these two positions. In order to compare the measurements to Mie theory, the polarization state must be included in the Mie calculation. This is not mentioned in the paper.*

*The scattered light will also be polarized and this needs to be addressed as well. For example, the detectors may detect different polarizations of light slightly differently. A flat-field calibration across the CCDs is also needed to verify that different angles of incident light to the detector are measured with equal sensitivity.*

**Response:** This is a good question. We do consider the polarization state of the laser in this work. A quarter-wave plate was inserted above the laser to change the polarization from linear to circular. During the exposure time of the image, the circular-polarization effects mimic unpolarised light. We added the explanation of the state of polarization into the manuscript at P2L9 as:

"To change the polarization state of the laser from linear to circular, a quarter-wave plate was mounted in front of the laser emitter. During the exposure time (few minutes) of the image, the circular-polarization light can be assumed as unpolarized."

The sensitivity of each pixel is influenced by CCD and lens. The CCD sensor we used is KAF-8300 whose quantum efficiency is about 55% at 532nm. The linear error of the CCD is about 10% according to the manual.

[Figure]

Fig.1 Modulation transfer function of Sigma 10mm F2.8 fisheye lens

The modulation transfer function of the lens we used is shown in Fig.1 (http://www.sigma-photo.co.jp/english/lens/wide/10_28/#/data). In this figure, the solid line show the performance of lens along the diagonal line. X axis shows the distance from the centre of the image. Because the diagonal line of the CCD is 22.5 mm length, the distance between the central pixels and the corner is about 11.3 mm. In this range the performance of lens is almost constant. Some parameters about the CCD and lens are added into the manuscript at P3L18 and P3L23 as:

"The quantum efficiency of the CCD is about 55% at 532nm, while the linearity error is about 10%."

"The modulation transfer function of the lens shows that, according to the size of the CCD sensor, the difference of the sensitivities from the centre to the corner is less than 5% (http://www.sigma-photo.co.jp/english/lens/wide/10_28/#/data)."

The detailed error analysis has been added as Section 2.2.3.

*2. How can the authors be sure that they are not detecting scattered light from multiple points along the laser into their CCD detectors? Although this is mentioned in the text, it is not clear that this is true.*

*In Figure 1, the authors show the setup, including scattering angle into the detector. But the aerosol is not constrained in any way along the path of the laser. Therefore, light scattering from a particle near the laser and from a particle at a different point along the laser beam could strike the detector at the same location.*

*It would be highly useful to show some raw data from the two CCD detectors for one measurement to show the relationship between the two and how the data is spliced together. At what scattering angles do the two CCD measurements overlap?*

*How do the authors determine the scattering angle that corresponds to each pixel position on the CCD? Is there some sort of measurement of a known scatterer that could be used to calibrate the angle, such as PSL or ammonium sulfate?*

*In general, it would be a good idea to perform the measurement with a known compound to better determine the efficacy of the instrument.*

**Response:** Thanks a lot for the suggestions.

The CCD detectors are used to capture image of the laser beam. The Figure 1(b) in the revised manuscript has been added to show the image captured by CCDs. According to the principle of image formation by lenses, there is a one-to one correspondence between the image and the laser beam object (https://en.wikipedia.org/wiki/Lens_(optics)). The explanation has also been added at P4L2 as:

"With the mounted lens, there is a one-to-one correspondence between the image of the laser beam captured by CCDs and the laser beam object according to the principle of image formation by lenses."

According to your suggestion, more details about the geometric measuring process and the data merging have been added in Section 2.2.1 at P4L16 and P5L16 respectively, as:

"The data acquisition of CCD-LADS is to obtain the angle-resolved scattering signals from images captured by two independent CCD systems, and then merge the signals. Firstly, the CCD-LADS is set up as the Figure 1(a) shows. The geometric relationships among the CCDs, laser emitter and light trap are measured by tape. Then the scattering angle of laser in the image should be calibrated. Firstly, the direction of the CCD cameras are adjusted to make sure that the image of laser is go through the centre of the pixel arrays of CCD. By using a beam block to block the backscattering light into the CCD, the pixel related to the 90° scattering angle can be indicated on the calibrated image (Figure 1(d)). Because of the equisolid projection is used by the lens, the distance from a point on the image on the CCD to the centre of the pixels can be represented as $R = 2f \times \sin(\theta/2)$, where $\theta$ is the angle in rad between a point in the real world and the optical axis, which goes from the center of the image through the center of the lens, f is the focal length of the lens (Miyamoto, 1964). So the scattering angle which the centre of the image related to can be calculated by substituting the distance from the pixel related to the 90° scattering angle to the centre of pixels in the calibrated image into the equation of the equisolid projection. A one-to-one correspondence between the image of laser and the scattering angle can be calibrated by this method."

"When the angle-resolved signals from two CCDs are obtained, the change of signals with angles can be merged by following the steps below. Firstly, the minimum angle $\theta_1$ and maximum angle $\theta_2$ of the overlap angular region of signals from two CCDs are set as the boundary angle of data merging (shadow zone in Figure 3). $\theta_1$ and $\theta_2$ are always around 50° and 80°, respectively. In this region, a transform coefficient with scattering angles $T(\theta)$ is calculated,

$$T(\theta) = \frac{I_1(\theta)}{I_2(\theta)} \tag{2}$$

$I_1(\theta)$ is the signal with the scattering angle $\theta$ captured by the first CCD while $I_2(\theta)$ is that of the second CCD. The lifted signal $I_2'(\theta)$ can be calculated by multiplying $I_2(\theta)$ with the average of $T(\theta)$ (Figure 3). For the region where $\theta < \theta_1$ or $\theta > \theta_2$, the signal $I_1(\theta)$ or $I_2'(\theta)$ is used as the merged scattering signal $I(\theta)$, respectively. For the overlap region, a linear weighting average is done between $I_1(\theta)$ and $I_2'(\theta)$,

$$I(\theta) = \frac{\theta_2 - \theta}{\theta_2 - \theta_1} \times I_1(\theta) + \frac{\theta - \theta_1}{\theta_2 - \theta_1} \times I_2'(\theta).$$

Using the method above, the merged signals with scattering angles $I(\theta)$ can be estimated."

The CCD-LADS is designed as an open path system to measure the ambient aerosols directly. Using a known compound just like PSL to calibrate the system is difficult because there is not a cavity in this system.

*3. More detail is needed regarding many aspects of the study. There needs to be more information regarding how the Mie calculations were performed. What was the refractive index used? Were the size distributions from the SMPS?*

**Response:** Thanks for your suggestion. We have added more information about the Mie calculation we used at P8L19, as:

"Combining the particle number size distributions measured with SMPS/APS and the mass concentration of black carbon aerosols measured with AE51 which are shown in Figure 6 into a modified Mie-scattering model, the aerosol optical properties including the aerosol phase function could be modelled (Ma et al., 2011). In this study (both laboratory and field study), the refractive index used for black carbon component is 1.95-0.79i (Seinfeld and Pandis, 2006), and for non-absorbing component is $1.53-10^{-7}i$ (Wex et al., 2002). The mass ratio between two different mixing states (external or core-shell, which means the different mixing way of black carbon and non-absorbing aerosols) of black carbon aerosols is assumed to be 1:1 according to the result of Ma et al. (2012)."

*4. The instrument will measure scattered sunlight when used during daylight hours. Using a 532 nm filter in front of the cameras will block much of the scattered sunlight, but the sunlight near 532 nm will be scattered by the particles and will reach the detector. Was there any attempt to block the sun from reaching the particles? For example, could a dark curtain or sampling box be used?*

**Response:** Thanks for the suggestions. Because the scattering phase function of the ambient aerosols is supposed to be observed, the noise from the

sunlight is an obvious issue indeed. Because the signal to noise ratio is lower during the daytime, actually we just used the nocturnal data to analyse. This issue was mentioned at the last sentence of Section 2.1. According to your comment, a further sentence has been added at P4L12 as:

"Currently, the CCD-LADS system can just estimate the nocturnal aerosol scattering phase function."

*5. Fluctuating temperature in the ambient measurement will likely cause fluctuations in the laser alignment during the measurements. Also, temperature fluctuations could affect the CCD efficiencies. Were the CCD cameras electronically cooled or was the temperature stabilized in any way?*

**Response:** After the system deployed, the laser alignment would be checked in every image. It seemd that the fluctuating temperature do not influence the laser alignment during night time in our observations. The CCD camera has an air-cooling unit. We added the description about this unit and the related setting at P3L18 and P4L28 as:

"This camera has an air-cooling unit to control the temperature of CCD."

"At the beginning of the measurement, the CCDs are cooled down to -15°C to minimize the noise from dark current."

*6. The authors claim that a significant portion of the particles measured were biomass burning particles. These particles could absorb a significant fraction of light. Was the absorption considered in the Mie calculations or was only scattering considered?*

**Response:** Both scattering and absorption are considered in the Mie calculation. An aethalometer was used in the observations to measure the absorption coefficient. The detailed description about the parameter used in

the Mie calculation have been added in the manuscript. Please see the response to specific comment 3.

*7. Ambient gases, such as nitrogen dioxide and ozone could absorb a significant fraction of light. It would be useful on Page 4, Line 20, for example, to discuss more detail in the assumption that the transmittance of the laser is 1.*

**Response:** The maximum mixing ratio of $NO_2$ and $O_3$ are about 100ppb and 200ppb in the North China Plain, respectively (Xu et al., 2011). The absorption cross sections of these two gases on the 532nm wavelength are measured in the past studies (Dixon, 1940; Burrows et al., 1999). According to the parameters in these studies, the maximum absorption coefficient of $NO_2$ is about 20Mm$^{-1}$, which have the same order with the scattering coefficient of air molecules. The maximum absorption coefficient of $O_3$ is about 2Mm$^{-1}$ which can be ignored compared with the absorption of $NO_2$ on this wavelength. According to the area that the CCD-LADS covers, the extinction coefficient of air molecules which including the scattering coefficient and the absorption coefficient of all the ambient gases can lead to a transmittance of 99.94% at most. In some extreme pollution processes, the extinction coefficient of aerosols may be exceeded 2000 Mm$^{-1}$ which can lead to a transmittance of 96.85% (Ma et al., 2011). Considering this issue, a threshold of visibility has been added to evaluate if the assumption used or not. The sentences have been added at P6L4 as:

"Depend on the area that the CCD-LADS covers, the longest distance between CCD cameras and the laser beam is less than 8m. In this range, an assumption that $\tau_Z = \tau_R = 1$ can be established with a threshold that the visibility should be larger than 1.5km. The correlation between the visibility and extinction coefficient $k_{ex}$ can be expressed as $k_{ex} = 3/visibility(km)$ (Chen et al., 2012) which means that the

assumption can be established if $k_{ex}$ is smaller than 2km$^{-1}$. In some extreme pollution processes while both aerosols and polluted gases are in large quantities (Ma et al., 2011; Xu et al., 2011), the scattering and absorption of aerosols and gases (NO$_2$ (Dixon, 1940), O$_3$ (Burrows et al., 1999), etc.) may lead to a marvelous $k_{ex}$. If the $k_{ex}$ is more than 2km$^{-1}$, the assumption cannot be applied while the transmittance can calculated with the measurement of visibility."

*8. The entire discussion of the retrieval algorithm is not clear (Section 2.2.2). More detail is needed. Please describe exactly how the retrieval algorithm works. Why is the retrieval needed? You are measuring the phase function, correct? Why would you need to retrieve it?*

**Response:** Thanks for the suggestion. The direct measured parameter of the CCD-LADS is the scattering phase function of the ambient air mass including air molecules and aerosols. The retrieval algorithm is to obtain the scattering phase function of ambient aerosols. The difference between the aerosol phase function obtained with the retrieval algorithm and measured directly is shown in Figure 7. We rewrite the Section 2.2.2 as:

[revised manuscript text omitted]

*9. Error bars (uncertainties) of the measured phase functions are needed to better understand the overlap between the models and measurements (in Figure 4, for example).*

**Response:** We show the standard deviation of CCD-LADS results in the field measurements in the last version of manuscript. Following this comment, the error bars of the laboratory results have also been added. Please see Figure 7 in the revised manuscript and Section 3.1. Section 2.2.3 has been added to analyse the uncertainties in the measurement.

**Technical corrections:**

*In general, there are not enough citations to the literature in the introduction. For example, Bohren and Huffman should be cited for Mie theory. The IPCC report should be cited for the first sentence (Page 1, Line 20). There are more field measurements that used an integrating nephelometer. Overall, more citations are needed.*

**Response:** Thanks for the comment. More citations have been added including Bohren and Huffman (2008), IPCC assessment report 5 (Pachauri et al., 2014) and the nephelometer works (Heintzenberg et al., (1996) etc.) at P2L1, P1L21 and P1L22.

*Throughout the paper, "figure" should be capitalized as "Figure 1".*

**Response:** Thanks. It has been corrected.

*$N_0$ and T are both used as symbols for transform coefficient (Page 4, Lines 6 and 19). How are these different?*

**Response:** Thanks for your suggestion. These two parameters are independent. The name of $N_0$ have been changed to "calibration factor" to distinguish with T.

*Page 1, Line 24 should read "The aerosol phase function. . .".*

**Response:** Thanks. It has been corrected.

*Page 2, Line 5 should read "A plane mirror was placed at that point and reflected the scattering signals. . ."*

**Response:** Thanks. It has been corrected.

*Page 2, Line 7: Is "linearity" correct? If so, please explain.*

**Response:** Thanks for the comment. That word which is a typo has been deleted.

*Page 2, Line 28. "mainly" is not needed. More information is needed about the laser. The make, model, etc. is needed.*

**Response:** Thanks. "mainly" was used because the quarter-wave plate was also considered as a part of the emitting system. More information about the laser has been added at P3L9 as:

"The transverse mode is near TEM00. The M2 factor is less than 2.0 while the divergence of beam is less than 2.0mrad. The diameter at the aperture is 3.0mm."

*Page 3, Line 20 should read "The CCD-LADS system covers an area 12 m long and 1 m wide."*

**Response:** Thanks. It has been corrected.

*Page 3, Line 21. How is the background noise measured? It would be useful to show a plot of the signals measured during the daytime and nighttime. Also, why is the noise so much higher in the daytime? See the comment above regarding the filters above. I suspect that you have a large portion of ambient light being scattered by the particles. Depending on the angle of the sun, this could affect the ambient measurements.*

**Response:** Thanks for the suggestion again. Currently, the CCD-LADS system can just estimate the nocturnal aerosol scattering phase function. Please see the response to specific comment 4. During the night time, the background noise can be subtracted by the normal fitting of signals. This process has been added in Section 2.2.1 at P5L6 as:

"After image captured, the scattering light of the laser beam are separated from the background noise in the image as the follow steps. Firstly, the central axis of the scattering signals of laser beam is fitted in the

program (the red line shown in Figure 2(b)). Then the intensities of image on the perpendicular of this central axis (the blue line shown in Figure 2(b)) are fitted with a normal distribution,

$$f(x) = I_0 + I \times \frac{1}{\sqrt{2\pi}\sigma} exp\left(-\frac{(x-\mu)^2}{2\sigma^2}\right) \tag{1}$$

where $I_0$ is the intensity of the background noise, $I$ is the intensity of the scattering signal of the laser beam related to one scattering angle, $x$ represents the distance between the pixel on the perpendicular and on the central axis of the scattering signals, $\sigma$ and $\mu$ are the fitting parameters of the normal distribution. Combining with the calibrated one-to-one correspondence between the image of laser and the scattering angle, the angle-resolved scattering signals is obtained with the above steps of data acquisition."

*Page 4, Lines 8-13. More detail is needed about the linear weighting and the data processing in general.*

**Response:** Thanks. More details have been added and Section 2.2.1 have been rewritten. Please see the new version of the manuscript.

*Page 4, Line 30. The authors state that the hemispheric backscattering is half of the full scattering. This is true for Rayleigh scattering, but it depends on the polarization of the light. Bohren and Huffman, or a Rayleigh scattering calculation needs to be cited here.*

**Response:** The laser is seemed as unpolarised by using a quarter-wave plate. Please see the response to specific comment 1. The citation has been added at P6L22.

*Page 5, Lines 1-9. This section is very confusing. Was a separate measurement of the air molecule scattering measured by an integrating nephelometer during this study? More detail is needed. What is meant by "the percentage of air scattering", for example? See the comment above.*

**Response:** Thanks for the suggestion. The air molecule scattering is calculated by the formula of Rayleigh scattering with the measured temperature and relative humidity. "the percentage of air scattering" means the fraction of air scattering coefficient in the total scattering coefficient ($\frac{k_{sc-air}}{k_{sc-air}+k_{sc-aero}}$). "the percentage of air scattering" has been expanded as "the percentage of air scattering in total scattering" in the revised manuscript. The Section 2.2.2 has been rewritten according to your suggestion. Please see the response to the specific comment 8.

*Page 6, Line 13. What is the external/core-shell ratio of 1:1? What does this mean? 1:1 core and shell ratio by volume?*

**Response:** External state and core-shell state are different mixing states of black carbon (BC). The mixing state will influence the optical properties of aerosols. In the Mie calculation, the BC mass ratio of different mixing state is needed. The detailed descriptions of Mie calculation are added. Please see the response to specific comment 3.

*Page 6, Line 23. The explanation that the scattering abilities of aerosol particles and gas molecules match better in the back scatter than the forward scatter is not clear. In the phase function, the Rayleigh and Mie scattering generally match in the forward and reverse directions (0 and 180 degrees).*

**Response:** Thanks for your comment. Here the "scattering abilities" should be instead by "scattering coefficients". In general, the scattering coefficient of aerosol is much higher than the scattering coefficient of air molecule. Based on this background, the back-scattering coefficients of aerosol and air molecule are closer to each other than the forward-scattering coefficients

according to the different patterns of the phase function of Rayleigh and Mie scattering. The sentence has been rewritten at P9L5 as:

"The reason of this phenomenon is that the scattering coefficients of aerosols and air molecules are closer to each other for the backward scatter than for the forward scatter based on the background that the total scattering coefficient of aerosols is always much higher than of air molecules."

*Page 6, Line 25 and Figure 4. Biomass burning aerosol is not shown on the graph. Also, this needs a reference for the discussion about biomass burning size, absorption, etc.*

**Response:** The scattering phase function of the "biomass burning" aerosol from the Cloud-Aerosol Lidar and Infrared Pathfinder Satellite Observations (CALIPSO) aerosol products (Omar et al., 2009) has been added in Figure 7 in the revised manuscript. The description of the aerosol classification from CALIPSO and the typical characteristics of biomass burning aerosols have been added with the related reference at P8L27 and P9L9, as:

"The CALIPSO aerosol classifications are based on the cluster analysis of the Aerosol Robotic Network (AERONET) measurements to determine characteristic aerosol types (Omar et al., 2005)."

"Compared with the other aerosol types, the "biomass burning" aerosol represents a better absorption ability due to the larger percentage of black carbon aerosol and organic aerosol, and also a smaller effective diameter around 100nm (Omar et al., 2005; Rissler et al., 2006; Zhu et al., 2017)."

*Page 7, Line 2. The authors state that the difference "Is not obvious", but there is clearly a difference between the Mie model and the measurement in Figure 5. This is not clear.*

**Response:** We deleted the "Is not obvious" in the revised manuscript, rewritten it as "the average difference in the absolute values between these two p(θ) is less than 10%" at P9L18.

*Page 7, Line 13. Why was the instrument limited to 30-160 degrees because of the ambient conditions?*

**Response:** The mobile laboratory was deployed on the roof of a building. To compare with the other in-situ measurements, the CCD-LADS was also installed on the roof. But there were limited space for a standard CCD-LADS system. So we shorten the distance between the laser and the beam trap, which cause the limitation of the detection range.

*Page 7, Line 16. It is more correct to say that the particles will exhibit hygroscopic growth rather than "grow up". Also in Line 18, why was the data collected at RH >70% eliminated? How was 70% RH chosen? Many particles exhibit water uptake at lower relative humidities. Biomass burning aerosol is highly hygroscopic and shows continuous growth. 70% seems to be arbitrarily chosen.*

**Response:** Thanks. "Grow up" has been instead by "exhibit hygroscopic growth" at P10L1. We chose the 70% RH because the deliquescence relative humidity is always beyond 73% in this region in our previous study (Kuang et al., 2016). The ratio of aerosol light scattering coefficients at 70% RH and at dry condition is about 1.3 (Chen et al., 2014), which will not influence the result too much. If a 50% or 60% RH is chosen, the comparison between CCD-LADS measurements and Mie calculations may be better, but the data quantity will also decrease.

*Page 7, Line 18. It is better to say "eliminated" than "kicked off".*

**Response:** Thanks. It has been corrected.

*Page 7, Line 26. What is meant by better stability? Was the data less noisy? This is not clear.*

**Response:** Compared to the Aurora 4000 measurements, the deviation of the CCD-LADS measurements is significantly less obvious (Fig.2). That's the reason why we use "better stability" as an advantage of CCD-LADS.

[Figure]

Fig. 2 Comparison between aerosol phase function at 42° scattering angle obtained from CCD-LADS measurements, Aurora 4000 measurements and modelled with modified Mie model.

*Page 8, Line 18. What is meant by "steadier"?*

**Response:** "steadier" is as the same meaning as "better stability" that we explained in the last response.

*Figure 3: The data shown here was not collected by the CCD-LADS instrument that is the focus of this paper. Why is this data shown? Was it used in the Mie calculation to compare to the CCD-LADS measurements?*

**Response:** Yes. These characteristics are used in the Mie calculation and also the retrieval algorithm to determine the aerosol phase function with CCD-LADS measurements. The explanation of this figure has been added at P8L19 as:

"Combining the particle number size distributions measured with SMPS/APS and the mass concentration of black carbon aerosols measured with AE51 which are shown in Figure 6 into a modified Mie-scattering model, the aerosol optical properties including the aerosol phase function could be modelled (Ma et al., 2011)."

*add uncertainties on the phase function provided by the new system. For example, the relative angle between the laser beam and the optic axis of the camera might be not exactly as what you expected. Also, there is always ambient light influencing the signal. I suggest the author to add a new section discussing about all those possible uncertainty sources, and give an overall estimate of the measurement uncertainty.*

**Response:** Thanks for the referee's comments. Section 2.2.3 has been added to analyse the errors in the data acquisition and retrieval algorithm, as:

"Two types of uncertainties determine the error of the retrieved aerosol phase function: the measuring errors caused by the processes to obtain the angle-resolved signals, and an error introduced by the retrieval algorithm.

There are two sources of measuring errors in the data acquisition processes introduced in Section 2.2.1. Firstly, the measuring error of CCD used in the CCD-LADS is 10% according to the related manual. The relative difference between the fitted normal distribution introduced in equation (1) and the measured signal in the laboratory study is 8.8% $\pm 1.5\%$, which can also certify the 10% measuring error on $I$ introduced by the manual of CCD. Secondly, the measurement of the geometric relationship will lead to at most 5% relative error on scattering angle $\theta$ introduced by the resolution and accuracy of the used tools.

The relative errors on the merged angle-resolved signals $I(\theta)$ can be derived by applying a standard propagation of errors to equation (3) (Bevington and Robinson, 2003),

$$\left(\frac{\Delta I}{I}\right)^2 = F_{I_1}\left(\frac{\Delta I_1}{I_1}\right)^2 + F_{I_2'}\left(\frac{\Delta I_2'}{I_2'}\right)^2 + F_\theta\left(\frac{\Delta\theta}{\theta}\right)^2$$

(10)

where $\sigma$ symbol means the standard deviation of variables, $\frac{\Delta x}{x}$ is equal to the relative error of $x$ and the propagation factor $F_x$ are defined as $F_x = \left(\frac{x}{I}\frac{\partial I}{\partial x}\right)^2$. By substituting the relative errors and the average signals into equation (10), the uncertainties on $I(\theta)$ are calculated as a distribution with angular shown in Figure 5.

The uncertainties of the retrieval algorithm are introduced by the uncertainties of the input parameters. There are three groups of input parameters in the retrieval algorithm: merged angle-resolved signals, aerosol hemi-backscattering coefficient and temperature/pressure. The errors of the temperature and pressure are about 0.1K and 0.1hPa (Box and Steffen, 2001), respectively, which will lead to a 0.02% uncertainty on $k_{bsc-air}$. Combined the 10% uncertainties on the measured $k_{bsc-aero}$ (Heintzenberg et al., 2006), the uncertainty of $R_{air}$ can be calculated as 7% with the algorithm in Section 2.2.2. According to the algorithm shown in Figure 4, the uncertainty of the retrieved aerosol phase function are mainly dominated by the uncertainties of the merged signal shown in Figure 5, and also influenced by the uncertainty of $R_{air}$ in a way."

*3. the English language needs to be improved.*

**Response:** Thanks. The English has been improved in the revised manuscript.

**Minor comments:**

*Section 1: The background at the beginning of section 1 is very weak. I suggest the author to write a bit more about why the phase function is important.*

**Response:** Thanks. The importance of aerosol phase function has been added at P1L25 as:

"The aerosol phase function ($p(\theta)$) is defined to describe the angular distribution of the aerosol scattering intensity (Hulst, 1957). $p(\theta)$ is one of the important properties controlling aerosol contribution to radiation balance of atmosphere (Andrews et al., 2006). Some parameters such as asymmetry parameter and hemispheric backscatter fraction estimated from $p(\theta)$ are of great importance to the retrieval of remote sensing measurements and the simulation of atmospheric radiative transfer model (Muñoz et al., 2002)."

*In section 1 you listed many methods which can provide aerosol phase function. Are they widely applied? If not, why? What are the advantage and disadvantage of those methods? One can give very short comments on each method.*

**Response:** Thanks. The descriptions about the previous methods which can provide aerosol phase function have been added at P2L9, P2L13 and P2L18 as:

"In past years, different research groups have developed several versions of polar nephelometers to measure how the scattering intensities of aerosol particles, cloud droplets and ice crystals changes with scattering angle. Muñoz et al. (2001, 2010, 2011) mounted a photomultiplier tube (PMT) on a mechanical arm which can rotate around a point on the laser light path in the same plane with the laser beam to change the scattering angle of the signal captured by the PMT. Castagner and Bigio (2006, 2007) focused the light scattered at a single spot with different scattering angles to another single spot by using two parabolic reflectors next to the light path. A plane mirror was placed at that point and reflect the scattering signals with different angles to a PMT by rotation. **These two styles of instruments measured the angular distribution of scattering signals by using the rotational mechanism. This design will lead to an obvious uncertainty because the signals were not measured simultaneously.** Barkey et al. (2002, 2007) made the sample flow perpendicular and intersect with the light path. Then many PMTs were mounted around the point of intersection in the same plane with the laser beam to capture the scattering signal from different scattering angles. **The signals with different scattering angles were measured at the same time with this design, however, the angular resolution which is limited to larger than 8° per point is relatively low because the PMTs cannot be mounted too close to each other.** Curtis et al. (2007, 2008) used an ellipsoidal mirror to reflect the scattering light to a charge-coupled device (CCD) detector for the detection of aerosol phase function. By using CCD as detector, this method can offer a better angle and time resolution at a wider range of scattering angles than the other methods above. It just needs one detector and there is no need to move the detector during the measurement. **However, the structure of this design is too complicated to be used into field measurement.**"

*Last para in section 1: it is better to start with "in this paper, we propose. . .". Otherwise it seems you are still talking about previous studies.*

**Response:** Thanks. According to your suggestion, the first sentence of this para has been revised as "In this paper, a novel instrument named……".

*P3L17: the detective angle range can be expanded to 10-170.*

**Response:** Thanks. The sentence has been revised by following the suggestion.

*P3L20: how do you measure the direction of the two CCD cameras? How do you ensure that they are pointing to the right direction?*

**Response:** Thanks. The geometric relationships among the CCDs, laser and beam trap are measured with tape. The direction of CCDs are calibrated by using a beam block to indicate the 90° scattering point on the image. The detailed calibrated method has been added at P4L16 as:

"The data acquisition of CCD-LADS is to obtain the angle-resolved scattering signals from images captured by two independent CCD systems, and then merge the signals. Firstly, the CCD-LADS is set up as the Figure 1(a) shows. The geometric relationships among the CCDs, laser emitter and light trap are measured by tape. Then the scattering angle of laser in the image should be calibrated. Firstly, the direction of the CCD cameras are adjusted to make sure that the image of laser is go through the centre of the pixel arrays of CCD. By using a beam block to block the backscattering light into the CCD, the pixel related to the 90° scattering angle can be indicated on the calibrated image (Figure 1(d)). Because of the equisolid projection is used by the lens, the distance from a point on the image on the CCD to the centre of the pixels can be represented as $R = 2f \times \sin(\theta/2)$, where $\theta$ is the angle in rad between a point in the real world and the optical axis, which goes from the center of the image through the center of the lens, f is the focal length of the lens (Miyamoto, 1964). So the scattering angle which the centre of the image related to can be calculated

by substituting the distance from the pixel related to the 90° scattering angle to the centre of pixels in the calibrated image into the equation of the equisolid projection. A one-to-one correspondence between the image of laser and the scattering angle can be calibrated by this method."

*P4L1: what is the typical exposure time?*

**Response:** Thanks. The exposure time of these two CCDs that are always about 5-60 seconds are tuned with the maximum of the signal intensity changing. The description of the exposure time has been added at P4L28 as:

"Then a test image with a 10s exposure time is captured to fix the exposure time of measurement by evaluating the signal intensity of this image. Generally, the maximum of the signal intensity is tuned to about 214 because the limitation is 216. If the maximum increased to the limitation in an image, the exposure time will also be changed in the next image automatically. The exposure time of these two CCDs that are always about 5-60 seconds in the past observations should be in complete accord for the comparison."

*P4L2: I do not understand "the central axis of the signals from the scattering light is fitted in the program". Explain it in detail or just remove it if it is not important for audience to understand the system.*

**Response:** Thanks. A Figure 2 has been added to explain the noise subtraction process which including the step which this sentence want to explain. The central optical axis is indicated in Figure 2(b). The sentence has been rewritten at P5L7 as:

"Firstly, the central axis of the scattering signals of laser beam is fitted in the program (the red line shown in Figure 2(b))."

*Fig2: signal merging is missing in this flowchart.*

**Response:** Thanks. Signal merging has been added in the flowchart in Figure 4 in the revised manuscript.

*P4L21: in extreme cases, e.g. heavy haze or fog, what is the uncertainty of assuming a tau of 1? Maybe one can give a threshold of visibility above which the uncertainty is negligible.*

**Response:** Thanks for the suggestion. A threshold of visibility has been added for the decision if the transmittance can be assumed as 1 or not at P6L5. If the visibility is lower than the threshold, the transmittance can be estimated by using the measurements with nephelometer and aethalometer. See the added sentences about the threshold below,

"In this range, an assumption that $\tau_Z = \tau_R = 1$ can be established with a threshold that the visibility should be larger than 1.5km. The correlation between the visibility and extinction coefficient $k_{ex}$ can be expressed as $k_{ex} = 3/visibility(km)$ (Chen et al., 2012) which means that the assumption can be established if $k_{ex}$ is smaller than 2km$^{-1}$. In some extreme pollution processes while both aerosols and polluted gases are in large quantities (Ma et al., 2011; Xu et al., 2011), the scattering and absorption of aerosols and gases (NO$_2$ (Dixon, 1940), O$_3$ (Burrows et al., 1999), etc.) may lead to a marvelous $k_{ex}$. If the $k_{ex}$ is more than 2km$^{-1}$, the assumption cannot be applied while the transmittance can calculated with the measurement of visibility."

*Section 3: Mie calculation was used to simulate the phase function at dry condition. Can you give an estimate of the uncertainty of the simulation? And I suggest the author to mark the uncertainties of measurement and simulation as error bars.*

**Response:** Thanks. The uncertainty of the simulation with Mie model is about 30% (Ma et al., 2011) due to the uncertainties of the input parameters. The standard deviation of CCD-LADS measurement and Mie simulation

have already shown in Figure 9 in the revised manuscript. Following the comment, the error bars have been added in Figure 7 in the revised manuscript to indicate the uncertainties.

*P6L25: I did not see "biomass burning" in figure 4.*

[revised manuscript text omitted]